# Co-Culture of Halotolerant Bacteria to Produce Poly(3-hydroxybutyrate-co-3-hydroxyvalerate) Using Sewage Wastewater Substrate

**DOI:** 10.3390/polym14224963

**Published:** 2022-11-16

**Authors:** Naima Khan, Iftikhar Ali, Sumaira Mazhar, Sajida Munir, Rida Batool, Nazia Jamil

**Affiliations:** 1Institute of Microbiology and Molecular Genetics, University of the Punjab, Lahore 54590, Pakistan; 2Department of Life Science, School of Science, University of Management and Technology, Lahore 54770, Pakistan; 3Department of Biology, Garrison University, Lahore 54810, Pakistan; 4Department of Medical Lab Technology, NUR International University Lahore, Lahore 55150, Pakistan

**Keywords:** co-culture, halotolerant bacteria, poly(3-hydroxybutyrate-co-3-hydroxyvalerate), sewage wastewater (SWW), propionic acid, pentanol

## Abstract

The focus of the current study was the use of sewage wastewater to obtain PHA from a co-culture to produce a sustainable polymer. Two halotolerant bacteria, *Bacillus halotolerans* 14SM (MZ801771) and *Bacillus aryabhattai* WK31 (MT453992), were grown in a consortium to produce PHA. Sewage wastewater (SWW) was used to produce PHA, and glucose was used as a reference substrate to compare the growth and PHA production parameters. Both bacterial strains produced PHA in monoculture, but a copolymer was obtained when the co-cultures were used. The co-culture accumulated a maximum of 54% after 24 h of incubation in 10% SWW. The intracellular granules indicated the presence of nucleation sites for granule initiation. The average granule size was recorded to be 231 nm; micrographs also indicated the presence of extracellular polymers and granule-associated proteins. Fourier transform infrared spectroscopy (FTIR) analysis of the polymer produced by the consortium showed a significant peak at 1731 cm^−1^, representing the C=O group. FTIR also presented peaks in the region of 2800 cm^−1^ to 2900 cm^−1^, indicating C-C stretching. Proton nuclear magnetic resonance (^1^HNMR) of the pure polymer indicated chemical shifts resulting from the proton of hydroxy valerate and hydroxybutyrate, confirming the production of poly(3-hydroxybutyrate-co-3-hydroxy valerate) (P3HBV). A 3-(4,5-dimethylthiazol-2-yl)-2,5-diphenyl-2H-tetrazolium bromide (MTT) assay showed that the copolymer was biocompatible, even at a high concentration of 5000 µg mL^−1^. The results of this study show that bacterial strains WK31 and 14SM can be used to synthesize a copolymer of butyrate and valerate using the volatile fatty acids present in the SWW, such as propionic acid or pentanoic acid. P3HBV can also be used to provide an extracellular matrix for cell-line growth without causing any cytotoxic effects.

## 1. Introduction

The increasing demand for synthetic polymers has led to a global plastic pollution problem. It is impossible to completely dispose of plastic waste or convert it into other useful products [1]. Biopolymers represent a wide range of alternatives that offer a natural and degradable solution for plastic pollution [2]. Polyhydroxyalkanoates (PHA) are biopolymers produced by microorganisms, preferably bacteria. PHAs are biodegradable, owing to the presence of an inherent depolymerase gene, phaZ, in the genomic DNA of the producer bacteria [3]. PHAs are also widely applicable in medicine, owing to their biocompatibility and ability to mimic an extracellular matrix [4]. Depending on the type of monomers, these polymers exhibit a wide range of thermal properties similar to those of synthetic polymers [5]. PHAs belong to three different classes: short-chain-length (C3-C6), medium-chain-length (C6-C10), and long-chain-length (>C10) polymers [6]. Various bacteria, such as *Pseudomonas* sp.; *Bacillus* sp., *Ralstonia eutropha,* also known as *Cupriavidus necatar*; *Rhodococcus* sp.; halotolerant bacteria; etc., can polymerize PHAs [7,8,9,10]. Bacteria that produce PHAs have an inherent advantage over non-producers, as these polymers enable them to survive nutrient limitations [11]. When PHA producers encounter a nitrogen-limited and carbon-excessive environment, they start polymerizing the excess carbon into PHAs to use under conditions of carbon limitation for energy and nutrient production [12]. Halophilic bacteria can withstand osmotic shock with the help of PHA granules, as their presence decreases plasmolysis in the producers [13]. Moreover, PHA has been reported to scatter ultraviolet radiation from bacterial cells, thus protecting against many environmental stresses beyond their traditional storage function [14,15].

Halophilic bacteria offer a promising polymerization system for secondary cell metabolites, such as PHA. Their ability to tolerate high salinity prevents microbial contamination, reducing unnecessary nutrient competition. Halotolerant bacteria increase the intracellular osmotic pressure, enabling rapid and easy cell lysis, making PHA recovery cost-effective. Another advantage among many is that halotolerant bacteria utilize various sustainable substrates, achieving possible low-cost, large-scale PHA production [16]. Co-culture systems often prove beneficial for secondary metabolite production. In co-culture, one partner enhances fermentation productivity and carbon utilization and provides essential monomeric subunits to produce an improved product. Co-cultures offer a unique opportunity to produce co-polymers while increasing carbon utilization from sustainable resources [17].

Sewerage water contains a variety of carbon sources; owing to the complex nature of such organic carbon sources, monocultures are often unable to simplify and use them. Co-culture works symbiotically, whereby one strain facilitates complex substrates, whereas others effectively convert them into important metabolites, such as PHA [18].

This aim of this study was to understand the growth kinetics, PHA production ability, and structure of extracted PHA from two halotolerant bacteria grown in co-cultures using a sustainable carbon source, i.e., sewage wastewater. The use of renewable carbon sources enables the cost-effective production of PHA. The current study allowed us to understand the timeline of PHA production by halotolerant *Bacillus aryabhattai* WK31 and *Bacillus halotolerans* 14SM during the first 92 h of growth in a continuous growth system.

## 2. Materials and Methods

### 2.1. Bacterial Strains and Sewage Wastewater

Bacterial strains WK31 and 14SM were isolated from extreme environments: hot water spring soil and the Khewra salt mines. The bacteria were Gram-stained for preliminary identification. The 16 s ribosomal rRNA gene was subjected to ribotyping, and GenBank accession numbers were obtained. Both strains were able to tolerate and grow in the presence of 3 M NaCl in the modified PHA detection medium, which also contains other salts. Sewage wastewater was obtained from a local drain and treated minimally before use [19]. The SWW was boiled and filtered to remove any suspended solids and used in a ratio of 1:10 with the PHA detection medium. The final growth medium was supplemented with 1.5 M NaCl to limit the growth of non-halophilic bacteria. Wastewater analysis was performed by the Pakistan Council of Scientific and Industrial Research (PCSIR).

### 2.2. Growth Kinetics of the Monocultures in Glucose and SWW

Selected bacterial strains, NK14 and WK31, were cultivated individually and in co-cultures. Experiments were conducted in 500 mL Erlenmeyer flasks at 30 °C, pH 6.8 ± 2, and 150 rpm for 92 h for bacterial cultivation in 2% glucose and 1.5 M NaCl-supplemented PHA detection medium (PDM) [20]. The initial optical density of cultures was optimized at 0.1 and thoroughly monitored during incubation. The biomass was collected at regular intervals, and the PHA concentration was evaluated for direct extraction from the biomass. Similarly, the monocultures of both strains were cultivated in a 10% SWW medium for PHA production. With the exception of the carbon sources, all other growth parameters were the same.

### 2.3. Co-Culture of WK31 and 14SM for PHA Production

Both strains used in co-culture were added in a ratio of 1:1 (1 mL each strain) to achieve an optical density of 0.1 at the start of the experiment. SWW and PDM (salt concentration was kept at 1.5 M) were used in a 1:10 ratio as carbon sources. Growth kinetics were also compared using glucose as a substrate. The experiments were conducted in 500 mL Erlenmeyer flasks at 30 °C, pH 6.8 ± 2, and 150 rpm for 72 h. The bacterial strain co-cultures were conducted under the same growth conditions, except for the initial seed culture. The biomass was collected at regular intervals, and PHA concentration was evaluated for direct extraction from the biomass.

### 2.4. Transmission Electron Microscopy of the PHA-Producing Bacterial Cells

The co-culture was inoculated in PDA medium [21] for 24 h in the presence of 1.5 M NaCl. After incubation, the cells were harvested by centrifuging 1.5 mL of culture at 1000 rpm for 5 min. The supernatant was discarded, and cells were suspended in 1 mL of fixative. The fixed cells were then sectioned to view the intracellular PHA granule accumulation. TEM micrographs were analyzed using Image J software.

### 2.5. Biomass Collection and PHA Extraction

The biomass was collected at regular intervals by centrifuging the culture medium at 6000 rpm for 3 min. The cells were washed with autoclaved distilled water and lyophilized. The cells were left in the chamber for 5 h at −20 °C under 60 m torr pressure. The weight of the freeze-dried cells was determined, and the cells were then subjected to polymer extraction. The biomass was dispensed with 6% sodium hypochlorite for one hour at 150 rpm. Sodium hypochlorite treatment was used to digest the cell debris, leaving the insoluble polymer. The purification of the polymer was ensured by dissolving it in chloroform (20 mL solvent, 1 g of biomass) and filtering to remove any cellular debris. The chloroform was evaporated under a fume hood, and the mass of the polymer was recorded.

### 2.6. Chemical Characterization of the Extracted Polymer

The extracted polymer was characterized using several techniques, including FTIR and ^1^HNMR. The extracted PHA was processed into a KBr disc and analyzed by infrared spectroscopy in the 400 to 4000 cm^−1^ range using an Alpha (BRUKER) spectrophotometer [22]. PHA proton nuclear magnetic resonance (1HNMR) scans were recorded after suspending the PHA in high-purity deuterochloroform (CDCl3). The ^1^H NMR spectra of the sample were obtained at 400 MHz using a Bruker Advance 300 NMR spectrometer (Rheinstetten, Germany). Chemical-shift scale values were recorded in parts per million (ppm).

### 2.7. Cytotoxic Effect of PHA Synthesized Using Wastewater

Wastewater contains numerous organic and inorganic waste compounds that can directly impact human health. Therefore, the cytotoxic effect of the PHA polymer biosynthesized by using wastewater was determined for hazard identification and risk assessment against HCT116 colorectal cell lines.

### 2.8. 3-(4,5-dimethylthiazol-2-yl)-2,5-diphenyltetrazolium Bromide (MTT) Assay

One of the most common methods used to determine cell viability and cytotoxicity is the MTT reduction test. Active mitochondria in live cells convert MTT into an insoluble purple formazan molecule [23]. The MTT test was used to assess the percentage viability of the cells under appropriate conditions. With slight changes, cell viability was measured using 0.5 mg ml^−1^ 3-(4,5-dimethylthiazol-2-yl)-2,5-diphenyltetrazolium bromide (MTT) [24]. This test measures the conversion of MTT to dark blue formazan precipitation by succinate dehydrogenase in undamaged mitochondria of live cells. The culture medium was gently withdrawn after stress treatment with PHA (10, 5, and 1 mg/mL), and an MTT reagent (10 µL) was applied to each well. The culture plates were then wrapped in aluminum foil and incubated at 37 °C for 3–4 h. When the dye came into contact with living cells, it formed purple-colored formazan crystals. After removing the dye, the crystals were resuspended in a 100 µL solubilizing agent. The plate was placed on a shaker to speed up the dissolution of these crystals. A microtiter plate reader was used to spectrophotometrically determine the optical density of the dissolved solution at 570 nm. The OD of treated wells was divided by the OD of vehicle control wells to assess the relative cell viability.

## 3. Results

### 3.1. Monoculture of Selected Bacteria Sewage Wastewater as Substrates

Bacterial strains were found to be capable of producing polyhydroxyalkanoate under salt stress [25]. *Bacillus aryabhattai* WK31 (MT453992) and *Bacillus halotolerans* 14SM (MZ801771) were grown in 2% glucose and 0.3% ammonium chloride under nitrogen limitation. The WK31 accumulated 63% PHA after 24 h of growth with 1.247 gL^−1^ biomass. The accumulated PHA began to decline thereafter, and biomass continued to increase, reaching a maximum of 3.446 gL^−1^ after 92 h of growth with a 7% PHA, as shown in Figure 1. Strain WK31 produced 54% PHA in the first 24 h of growth using SWW as a substrate. The PHA percentage began to decline, with a steady increase in biomass, reaching a maximum of 3 gL^−1^ after 96 h of incubation, as shown in Figure 1. Wastewater analysis results are provided in the Appendix A.

Strain 14SM produced copious amounts of PHA when glucose was used as a substrate. The strain accumulated 78% PHA after 24 h of incubation, achieving a biomass (dry cell weight) of 1.7 gL^−1^, as shown in Figure 2. Owing to the consumption of the carbon substrate, a decline in PHA accumulation was observed, and a minimum amount of 13% PHA was recorded after 96 h of incubation. When SWW was used as a substrate for 14SM, the increase in biomass was slow. After 48 h of incubation, the bacteria accumulated 51% PHA, which began to decline with further incubation (Figure 2).

### 3.2. Co-Culture of WK31 and 14SM Using Sewage WW as a Substrate

Bacterial strains 14SM and WK31 were grown in a co-culture. Glucose-supplemented growth medium was used as a control against sewage-wastewater-supplemented media. The composition of SWW is listed in the Appendix A. The synergism of these two strains was studied to produce PHA in glucose and wastewater-supplemented media. The co-culture produced a maximum biomass of 4.062 gL^−1^ in glucose-supplemented media. The maximum PHA production by WW and glucose-supplemented bacterial co-cultures was 54 and 60%, respectively, after 24 h of incubation in a shaking flask. The maximum yield of PHA using SWW was recorded to be 0.5 gL^−1^ after 24 h of incubation, whereas that of glucose was reported to be 0.8 gL^−1,^ as shown in Figure 3.

### 3.3. Structural Analysis of Intracellular PHA

The intracellular distribution of the PHA granules grown on wastewater as a carbon source was determined by transmission electron microscopy (TEM). TEM can distort the morphology of the cells or granules during sectioning and further processing of the cells [26]. TEM images of the bacterial culture after 24 h of cultivation showed that most of the PHA granules were located near the cell membrane, with some granules also present in the center of the cell. Most of the granules were circular, and a few were ovoid. The distribution of PHA granules is shown in two morphologically different cells (Figure 4C). In some of the granules, an extra granular layer is apparent. The cells also show indicate the decay or depolymerization of PHA granules in action. The black arrow in the figure indicates the rupture of the cellular membrane, which could have occurred due to the processing of the cells for TEM or cell lysis. The mean size of the PHA granules was calculated using Image J software. The mean granule size (MGS) in the rod-shaped cell shown in Figure 4C is 114 nm, whereas that in the coccobacilli cell is s231 nm. The granules have associated depolymerase proteins, which help to depolymerize the polymers and other granule-associated proteins. The blue arrow in the figure indicates a distinct extracellular EPS layer. The dark stained area indicated by the gray arrow in the micrographs shows the nucleation site for granule initiation [27]. Figure 4A,B represent the cellular morphology of the strain WK31 and 14SM at 1000X. The cells of the WK31 were Rhodococcus, while that of 14SM were streptobacilli. The TEM shows the presence of both bacterial strains in the co-culture, producing polymer simultaneously.

### 3.4. Structural Elucidation of the Extracted PHA

The FTIR spectrum of the PHA produced by co-culture indicates the presence of a carbonyl (C=O) functional group at 1732 cm^−1^. The spectrum also shows a transmittance at 2856 cm^−1^ and 2919 cm^−1^, representing an alkyl group stretching specific to mcl-PHAs as indicated in Figure 5 [27]. The spectrum also indicates multiple types of stretching for the alkyl group between 1200 cm^−1^ and 1500 cm^−1^. NMR images of the extracted and purified PHA were captured. As shown in Figure 6, the chemical shift at 0.85 could be attributed to protons connected to the methyl group of hydroxy valerate (CH3, C5). The peak at 1.65 results from the chemical shift of protons associated with hydroxybutyrate (HB, C4). According to proton NMR spectra, the PHA extracted from the co-culture of 14SM and WK31 was a copolymer of P3HBV [28]. The observed chemical shifts are similar to those previously reported in the literature and indicate that the major component of the polymer is hydroxyvalerate, as mentioned in Table 1 [29,30,31].

### 3.5. Biocompatibility of PHA Produced Using Wastewater and Open Mixed Culture

The biocompatibility of the extracted copolymer was evaluated in vitro at concentrations of 1000 µgml^−1^, 5000 µgml^−1^, and 10,000 µgml^−1^. The percentage cell viability of cell line HCT116 was 92% in the presence of 1000 µgml^−1^ PHA; however, it decreased at higher concentrations of PHA (60% at 5000 µgml^−1^ PHA and 58% at 10,000 µgml^−1^ PHA), as shown in Figure 7. mcl-PHA extracted from open mixed culture using wastewater as a carbon source was compatible with the cell lines and did not affect their viability. However, increasing the concentration to ≤5000 µgml^−1^ decreased the cell viability.

## 4. Discussion

PHB synthesis occurs through acetoacetyl-CoA polymerization, which is the result of glycolysis or the metabolism of sugars that enter glycolysis at various stages. Fatty acid degradation also contributes to the formation of volatile fatty acids and alkyl alcohols, such as propanol and pentanol, which are converted to monomers, such as valeric acid. Hydroxybutyric acid and valeric acid then combine to form a poly(3-hydroxybutyrate-co-3-hydroxyvalerate) [32].

Co-culture of bacteria synergistically utilizes complex substrates, such as agricultural waste, wastewater, molasses, etc. [33]. The breakdown of complex substrates by one partner makes the nutrients available for others and enhances the growth kinetics and production of useful products compared to monocultures [32]. In the current study, we selected two halotolerant bacterial strains—*Bacillus halotolerans* 14SM and *Bacillus aryabhattai* WK31—for PHA accumulation. Both selected bacteria were able to accumulate PHA in monocultures. However, structural analysis of the PHA showed that it was an scl-PHB with limited application. Sewage wastewater contains numerous substrates, such as sugars, lipids, and volatile fatty acids. SWW has been reported to produce 46 wt %PHA in mixed culture [34]. The diversity of substrates provides various monomers for synthesis of copolymers by the bacteria [35]. Volatile fatty acids, such as valeric acid, can be incorporated into PHB, making it less crystalline and improving its thermal properties [36]. Morgan-Sagastume et al. reported a 34% increase in poly(3-hydroxybutyrate-co-3-hydroxyvalerate) production in large-scale fermentation of volatile fatty acids [37]. When the selected bacteria were grown in a consortium, they produced an improved PHA copolymer, poly(3-hydroxybutyrate-co-3-hydroxyvalerate) (P3HBV), as confirmed by FTIR and proton NMR results. This polymer has a wide range of applications and is minimally crystalline, making it suitable to replace synthetic plastics. Glucose was used as a reference substrate for sewage wastewater. As it is the simplest sugar, the bacteria utilized glucose optimally, producing maximum PHA after 24 h of growth. The substrate was limited; therefore, after 24 h of growth, the bacteria used the intracellular PHA to increase the biomass. As indicated in Figure 3, the PHA concentration declined as the biomass increased, suggesting that the bacteria utilized the PHA to increase the biomass. After 24 h, the PHA yield declined, reaching a negligible amount. The decline in PHA synthesis shows that bacteria started utilizing the phasins and other enzymes exclusively involved in PHA synthesis to increase the biomass. Bacteria utilize excess carbon for PHA accumulation, but once the concentration of carbon is reduced, PHA production declines, and the stored PHA is depolymerized to increase the cell weight. Studies have demonstrated that the production of phasins, such as phaP, is associated with the production of or presence of PHB in the cells [38]. In this study, the continuous-batch technique was used, although a fed-batch technique and feeding of the culture with a carbon source at regular intervals may be optimal. TEM images of the bacterial cells showed the presence of two distinct cell types similar to 14SM and WK31, indicating that both cells were synergistic and did not produce any antimicrobial compounds to limit the growth of the other.

Wastewater contains many Gram-positive and Gram-negative bacterial species; even after sterilization, some toxins may exist. The copolymer was found to be biocompatible, as the cells were viable even at a high concentration of P3HBV (1000 µgmL^−1^); however, when the concentration of the P3HBV was further increased, the viability decreased by 30%, which could be attributed to either the increased concentrations of P3HBV or the increased the concentration surface-associated proteins or toxins, which had a negative effect on cell viability. PHAs have been reported to increase cellular growth, as they can be used to mimic intracellular metrics, providing a cellular network for the growing cells. Cell growth on PHA films has been reported to achieve improved growth and viability compared to other more plane surfaces, such as polyethylene plates or tubes [39]. The mechanical properties of PHA meet the requirements for soft tissue scaffolds, contributing to the growth and development of cell lines without causing any cytotoxic effects [40].

Figure 8 shows the butanoate metabolism in *Bacillus* sp. extracted from the Kyoto Encyclopedia of Genes and Genomes (KEGG). The figure shows the exchange of metabolic products and the use of various substrates for the biosynthesis of PHA. As indicated by highlighted areas, glucose, fatty acid, methyl acetate, propane, and butane from different metabolic pathways can contribute to the biosynthesis of various monomers for diverse polymers [41]. This figure is an oversimplified version of the PHA metabolic pathway, as the number and class of genes involved in the monomeric synthesis of PHA can vary, depending upon the substrate and the bacterial strains. In the current study, transmission electron microscopy images showed that both strains produced polymers, suggesting that both strains produce a similar polymer individually or simultaneously produce different polymers. In the current study, we did not focus on the genetics behind the metabolic interchange between the two strains. Detailed enzymatic and genomic analyses are required to understand the relationship between these two microbes. PHB synthesis occurs through acetoacetyl-CoA polymerization, which results from glycolysis or the metabolism of sugars that enter glycolysis at various stages. Fatty acid degradation also forms volatile fatty acids and alkyl alcohols, such as propanol and pentanol, which are converted to monomers, such as valeric acid. Hydroxybutyric acid and valeric acid then combine to form a poly(3-hydroxybutyrate-co-3-hydroxyvalerate) [42].

## 5. Conclusions

In this study, untreated sewage wastewater was used as a carbon source by two halotolerant bacterial strains, *Bacillus halotolerans* 14SM and *Bacillus aryabhattai* WK3. The wastewater contained a low carbon-to-nitrogen ratio, favoring PHA accumulation. Propionic acid was present in the SWW (53 mg mL^−1^), as determined by HPLC analysis. The presence of volatile fatty acids serves as a precursor for various monomeric units of PHA. The PHA extracted from the co-culture was poly(3-hydroxybutyrate-co-3-hydroxyvalerate). Neither of the strains were able to synthesize the co-polymers individually, showing that a consortium of bacteria can potentially provide improved compounds using a complex substrate, such as sewage wastewater. The reported results confirm that SWW can provide an essential environment for the synthesis of diverse types of PHA.

## Figures and Tables

**Figure 1 polymers-14-04963-f001:**
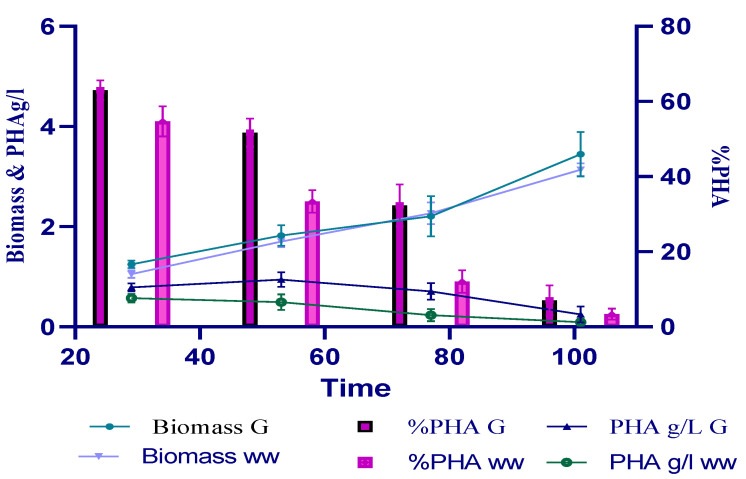
Production of PHA in monoculture. *Bacillus aryabhattai* WK31 bacteria were grown on 2% glucose and provided carbon access and a nitrogen-limited medium of 0.2 g ammonium chloride (2:0.2) in an PHA detection medium (PDM).

**Figure 2 polymers-14-04963-f002:**
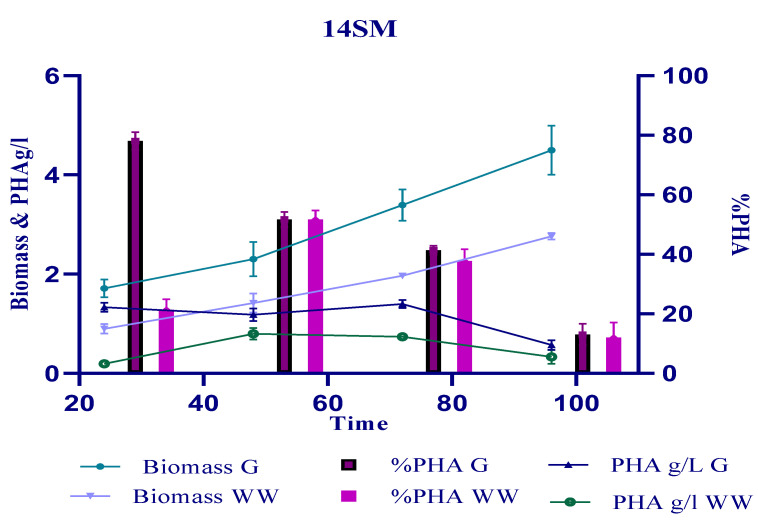
Production of PHA in the monoculture of *Bacillus halotolerans* 14SM using 2% glucose and 0.2% ammonium chloride.

**Figure 3 polymers-14-04963-f003:**
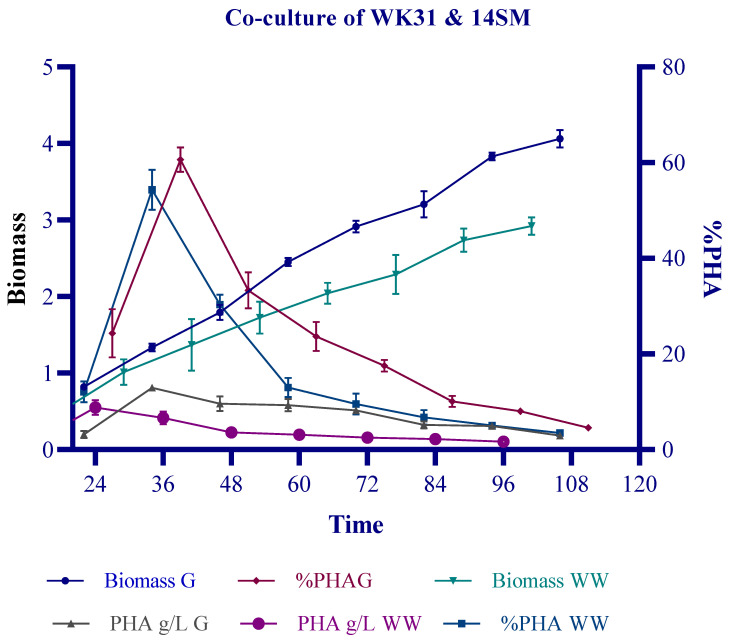
Co-culture in SWW and reference carbon source glucose: a comparative analysis of the PHA production potential of co-cultures of WK31 and 14SM under control (PDM-supplemented with 2% glucose) and experimental condition (1:1) PDM and wastewater.

**Figure 4 polymers-14-04963-f004:**
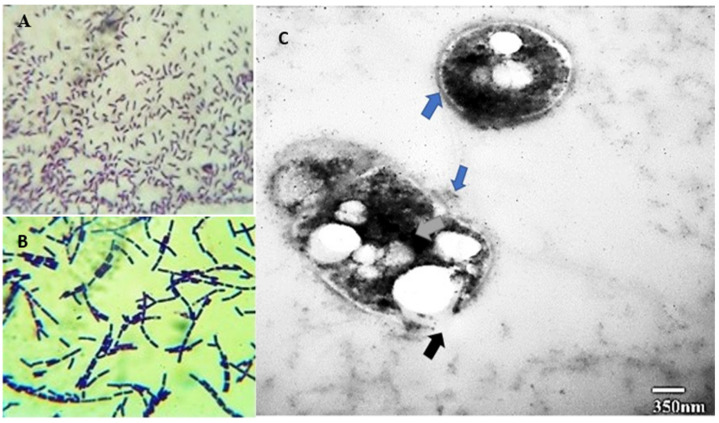
Granule formation in co-culture at 350 nm when grown in the presence of WW after 48 h of growth. (**A**,**B**) are micrographs of coccobacilli WK31 and bacilli 14SM, respectively at 1000×. (**C**) shows the presence of intracellular granules at 350 nm.

**Figure 5 polymers-14-04963-f005:**
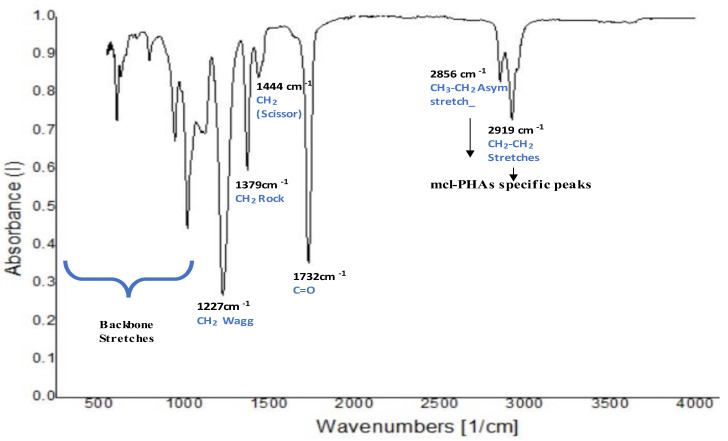
Fourier transform infrared spectroscopy of the PHA produced by co-culture of WK31 and 14SM using wastewater as a carbon source. The spectrum shows a distinct peak at 1732 cm^−1^, which indicates a C=O functional group, whereas the peaks in the region of 2800 to 2900 cm^−1^ represent C-C stretching.

**Figure 6 polymers-14-04963-f006:**
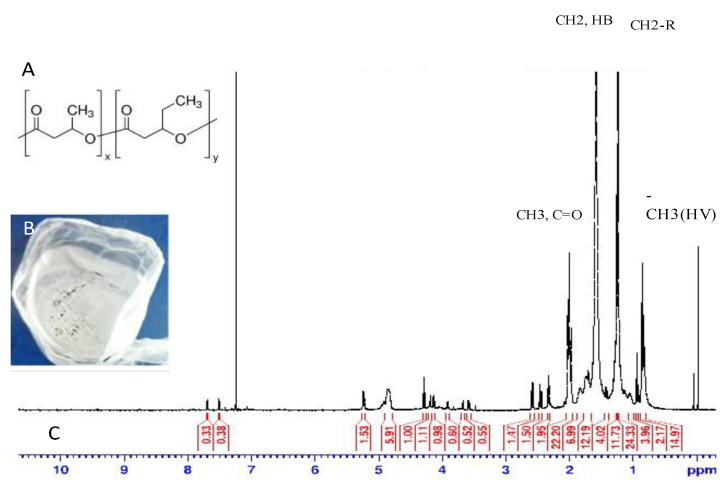
Proton NMR spectrum of extracted PHA from co-culture of WK31 and 14SM at 500 MHz using 1:1 WW and PDM as growth medium. (**A**) Chemical structure of the extracted polymer; (**B**) the extracted polymer; (**C**) proton NMR peaks of poly(3-hydroxybutyrate-co-3-hydroxyvalerate) (P3HBV).

**Figure 7 polymers-14-04963-f007:**
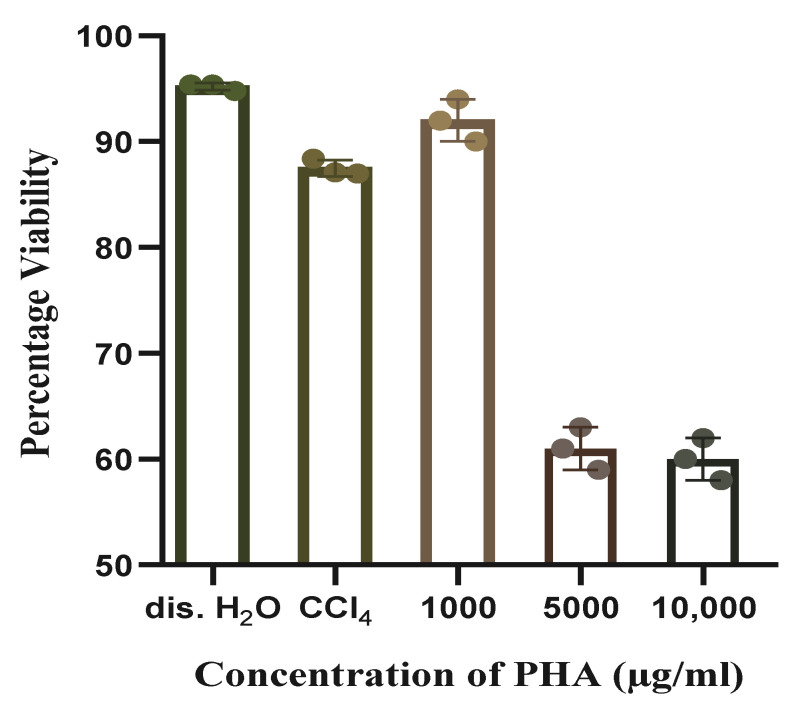
Percentage viability of cancer cell line HCT116 against varying concentrations of PHA obtained from a microbial consortium using wastewater as a carbon source. The copolymer was dissolved in chloroform; therefore, it was used as a control, along with autoclaved distilled water.

**Figure 8 polymers-14-04963-f008:**
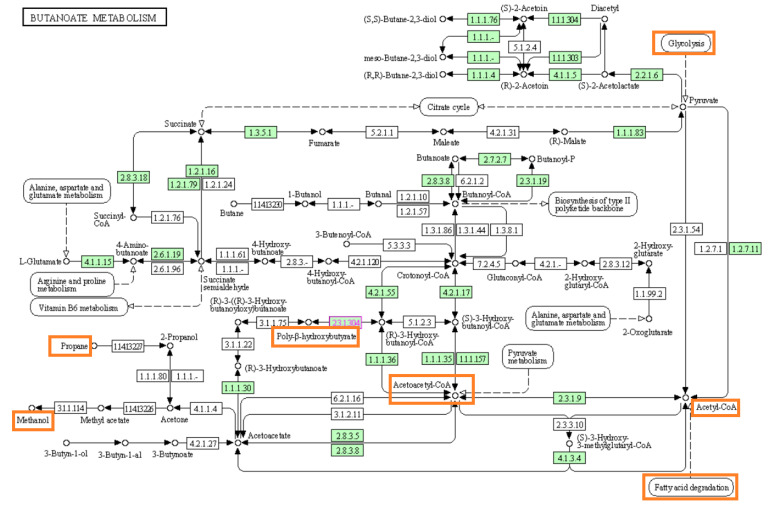
The metabolic pathway involved in the biosynthesis of polyhydroxybutyrate; the incorporation of alkyl alcohols leads to the synthesis of various monomers, such as valeric acid, for co-polymer production (extracted from KEEG).

**Table 1 polymers-14-04963-t001:** Chemical shifts (mean values) of ^1^HNMR and assignments of chemical groups of P3HBV.

^1^HNMR (ppm)	Integrated Signals	Diad/Triad	Chemical Group	
0.846	14.97	VV (t)	HV5	CH3
0.941	2.77	VV	HV5	CH3
1.248	11	BV(d)	HB4	CH3
1.582	12.20	VV*V	HV4	CH2
2.330	1.9	BV	HB2	CH2
2.452	1.81	B*V	HB2	CH2
2.584	1.47	BV*V	HB2	CH2
4.849	3.91	BB	HV3	CH
5.236	1.33	V*B	HB3	CH

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
