# Peer review of "Co-Culture of Halotolerant Bacteria to Produce Poly(3-hydroxybutyrate-co-3-hydroxyvalerate) Using Sewage Wastewater Substrate"

_polymers, 2022, doi:10.3390/polym14224963_

Round 1

Reviewer 1 Report

The manuscript by Khan and co-authors present the synthesis of different PHA (homo or copolymer) by a bacterial consortia. Thea authors demonstrate the possibility to use volatile fatty acids present in sewage wastewater were used as carbon source, as well as the biocompatibility of the obtained biopolymers.

General comments:

a) Standardize the units to 1/cm or cm-1, μg/ml or μg.mL-1

b) Materials and methods: include data on the used culture media, concentrations, volumes, information on how CDW and other parameters were analysed.

c) Figures quality, including captions, is really poor. Please improve and standardize.

Specific comments:

1) L 15. Please standardize to 'copolymer' - L 25 and throughout the text.

2) L 23. According to IUPAC, the correct nomenclature is poly(3-hydroxybutyrate-co-3-hydroxyvalerate) - please correct throughout the text.

3) L 26. Propionic acid and pentanoic acid?

4) L 36. A more adequate reference for PHA degradation is https://doi.org/10.1146/annurev.micro.56.012302.160838 Please check the traditional PHA literature, including 'The Handbook of Polyhydroxyalkanoates' https://www.taylorfrancis.com/books/edit/10.1201/9780429296611/handbook-polyhydroxyalkanoates-martin-koller

5) L 43. Ralstonia eutropha is also known as Cupriavidus necator - please include this information.

6) L 45. Please check other advantages on accumulating PHA https://doi.org/10.1016/j.nbt.2018.10.005; https://doi.org/10.1007/s00253-018-8760-8 ; https://doi.org/10.1016/j.biotechadv.2017.12.006

7) L 46-48. 10.1128/mr.54.4.450-472.1990 is the adequate reference here.

8) L 76-79. This sentence should be moved to the results section. If this is not part of your results, move it to the introduction section.

9) L 81. Please include the full composition of the used medium for seed cultures, PHA production etc.

10) L 87. Define PDM.

11) L 90. w/w ?

12) L 96. Specify the working volume.

13) L 115. Which cell line was used in this test?

14) L 129. Specify MTT reagent concentration.

15) L 140-141. The methodology to determine CDW and PHA amount and composition was not mentioned in the materials and methods section. Please clarify since this is essential information.

16) L 143. What is the limiting nutrient in this experiment? Did you measure nitrogen, phosphate or other nutrients concentration?

17) L 166. Again, clarify what is the nutritional limitation, include data on its concentration during the cultivation, etc.

18) L 169-171. Please characterize your carbon source - which acids were present? In which concentration?

19) L 170. Please determine the yield (g/g) of PHA from carbon sources and compare to the maximum theoretical yield. This is an essential information to determine how your bioprocess could be improved.

20) L 179. Please clarify why TEM was used to observe PHA granules. What information could you obtain from this analysis?

21) L 214-215. Do you have this analysis of the individual cultures? Could it be a blend and not a copolymer?

22) L 229. Controls are missing. Please include untreated and DMSO treated samples.

23) L 231. Which solvent was used to dissolve you PHA?

24) L 234. Could you detect the presence of solvents used for PHA extraction? Can these solvents affect your experiment?

25) L 239. Controls are missing.

26) L 254. Include in your discussion other studies using sewage wastewater for PHA production. Compare the obtained values and consider including a comparative table. Please check https://doi.org/10.2166/wst.2013.643

; https://pubs.acs.org/doi/10.1021/acs.iecr.9b01831 ; https://doi.org/10.1016/j.biteb.2021.100783 ; https://doi.org/10.3389/fbioe.2021.628719

27) L 255. Correct nomenclature according to IUPAC.

28) L 261. PHA was used to increase or to sustain biomass? Please indicate in your figures carbon source exhaustion, limiting nutrient concentration and exhaustion.

29) L 275. Did you determine mechanical properties of the produced PHA?

30) L 283-288. This information should be presented and discussed in the first part of the discussion.

31) L 284. Please elaborate more on the metabolic pathways to produce 3HA monomers. Including a figure from your substrates to monomers would be ideal.

32) Figure 8 was extracted from KEEG, please clarify this is not an original scheme.

33) L 291. Clarify the metabolic interchange in the proposed consortium.

Author Response

The changes suggested by the reviewer have been made. 

Please find the attached word file for detailed reviews. 

Reviewer 2 Report

The studies carried out by the authors of this paper involve the investigation of the formation method of the biopolymer, PHA, using microbial cultures and subsequently the identification and characterization of the obtained compound.

Weaknesses:

- Abbreviations appear in the Abstract and are not defined before.

- The legends of the figures must indicate the parameters being discussed, but details must be given in the text (for example Fig 1 and 2)

- Figure 4: Attention! The text talks about figure 6.

- Figure 8: Reference is also made to literature 31, but more comments must be made. A reader who is not familiar with the field will need support in understanding certain aspects.

- The conclusions are very evasive, they require great improvements.

Author Response

The authors have made the changes suggested by the reviewer.

Please find the attached word file for detailed answers. 

Round 2

Reviewer 1 Report

A few points must be clarified:

L43. Cupriavidus necator please correct

L 132. If you have NMR spectra, the 3HB and 3HV molar fractions of your polymers can be calculated. Please include this information to increase publication quality (check https://pubs.acs.org/doi/pdf/10.1021/acs.biomac.0c00826 for more information). This kind of information is essential on Polymer manuscripts.

L 163. If nitrogen limitation was used as a condition for PHA accumulation, how do you explain biomass increase after 24h?

L 194. Figure 3. It would be ideal to include information on nitrogen concentration during the cultivation, and to indicate the cultivation time point when limitation was achieved.

L 320-323. This sentence is not clear. Please clarify what you mean by viruses producing PHA.

Author Response

L43. Cupriavidus necator please correct: corrected

L 132. If you have NMR spectra, the 3HB and 3HV molar fractions of your polymers can be calculated. Please include this information to increase publication quality (check https://pubs.acs.org/doi/pdf/10.1021/acs.biomac.0c00826 for more information). This kind of information is essential on Polymer manuscripts. We have the data from proton NMR and a table has been added to further explain the data. 

L 163. If nitrogen limitation was used as a condition for PHA accumulation, how do you explain biomass increase after 24h?

L293-301 (reference added) After 24 hours, the PHA yield declined, reaching a negligible amount. The decline in PHA synthesis shows that bacteria started utilizing the phasins and other enzymes exclusively involved in PHA synthesis to increase the biomass. Bacteria utilize excess carbon for PHA accumulation, but once the concentration of carbon reduces, the PHA production declines, and the stored PHA is depolymerized to increase the cell weight. Studies have demonstrated that the production of phasins such as phaP is associated with the production of or presence of PHB in the cells.  In this study, the continuous-batch technique was used but using a fed-batch technique and feeding the culture with a carbon source after regular intervals would be wise.

L 194. Figure 3. It would be ideal for including information on nitrogen concentration during the cultivation and to indicate the cultivation time point when the limitation was achieved. At the start of the experiment, the ratio of carbon and nitrogen was kept at 0.2; for this experiment, the authors did not determine the nitrogen concentration. It was supposed that as the biomass increased, the concentration of carbon and nitrogen might have decreased, but their ratio remained comparably similar.  

L 320-323. This sentence is not clear. Please clarify what you mean by viruses producing PHA. The correction was made.